# Identification and Validation of PLOD2 as an Adverse Prognostic Biomarker for Oral Squamous Cell Carcinoma

**DOI:** 10.3390/biom11121842

**Published:** 2021-12-07

**Authors:** Yawei Sun, Shuai Wang, Xingwei Zhang, Zhuhao Wu, Zihui Li, Zhuang Ding, Xiaofeng Huang, Sheng Chen, Yue Jing, Xiaoxin Zhang, Liang Ding, Yuxian Song, Guowen Sun, Yanhong Ni

**Affiliations:** 1Central Laboratory of Stomatology, Nanjing Stomatological Hospital, Medical School of Nanjing University, Nanjing 210008, China; mf1935152@smail.nju.edu.cn (Y.S.); ws0826@outlook.com (S.W.); mf20350208@smail.nju.edu.cn (X.Z.); mf20350198@smail.nju.edu.cn (Z.W.); mf21350182@small.just.edu.cn (Z.L.); mg1935062@smail.nju.edu.cn (Z.D.); jingyuenju@outlook.com (Y.J.); sissi.117@hotmail.com (X.Z.); damoyu90@outlook.com (L.D.); songyuxiannju@outlook.com (Y.S.); 2Department of Oral Pathology, Nanjing Stomatological Hospital, Medical School of Nanjing University, Nanjing 210008, China; huangxiaofengnju@outlook.com (X.H.); chenshengnju@outlook.com (S.C.); 3Department of Oral and Maxillofacial Surgery, Nanjing Stomatological Hospital, Medical School of Nanjing University, Nanjing 210008, China

**Keywords:** PLOD2, oral squamous cell carcinoma, prognosis, metabolomics, epithelial–mesenchymal transformation

## Abstract

Background: Procollagen-lysine, 2-oxoglutarate 5-dioxygenase 2 (PLOD2), a key enzyme that catalyzes the hydroxylation of lysine, plays a crucial role in the progression of several solid tumors. However, its spatial expression profile and prognostic significance in oral squamous cell carcinoma (OSCC) have not been revealed. Materials: Mass spectrometry was used to explore amino acid perturbations between OSCC tumor tissues and paired normal tissues of 28 patients. Then, PLOD2 mRNA and protein levels were assessed using several public databases and 18 pairs of OSCC patients’ tissues. Additionally, PLOD2 spatial expression profiles were investigated in 100 OSCC patients by immunohistochemistry and its diagnostic and prognostic values were also evaluated. Lastly, gene set enrichment analysis (GSEA) was used to investigate the potential functions of PLOD2 in OSCC. Results: Lysine was significantly elevated in OSCC tissues and could effectively distinguish tumor from normal tissues (AUC = 0.859, *p* = 0.0035). PLOD2 mRNA and protein levels were highly increased in tumor tissues of head and neck squamous cell carcinoma (HNSCC) (*p* < 0.001) and OSCC compared with those in nontumor tissues (*p* < 0.001). Histopathologically, PLOD2 was ubiquitously expressed in tumor cells (TCs) and fibroblast-like cells (FLCs) of OSCC patients but absent in tumor-infiltrating lymphocytes (TILs). Patients with highly expressed PLOD2 in TCs (PLOD2^TCs^) and FLCs (PLOD2^FLCs^) showed poor differentiation, a worse pattern of invasion (WPOI) and more lymph node metastasis (LNM), contributing to higher postoperative metastasis risk and poor survival time. However, PLOD2^FLCs^ rather than PLOD2^TCs^ was an independent risk factor for survival outcomes in OSCC patients. Molecularly, GSEA demonstrated highly expressed PLOD2 was mainly enriched in epithelial–mesenchymal transformation (EMT), TGF-beta signaling and hypoxia pathway, which are associated with poor clinical outcomes of OSCC patients. Conclusions: PLOD2 was a poor prognostic biomarker for OSCC patients and may affect the metastasis of OSCC through EMT pathway. These findings might shed novel sights for future research in PLOD2 targeted OSCC therapy.

## 1. Introduction

Oral squamous cell carcinoma (OSCC) is the most common type of epithelial malignant tumor in the head and neck with 300,400 new cases and 145,400 deaths occurred every year around the world [1]. Despite the new concept and methods of treatments have been improving in recent years, the five-year survival rate of OSCC patients is still kept at 50–60% due to the inadequate early diagnostic and prognostic markers [2]. Therefore, there is a pressing need for more effective diagnostic and prognostic biomarkers.

Reprogramming of metabolism is a hallmark of malignant tumor, and metabolites have proved to be the ideal tumor biomarkers [3,4]. Our previous study has reported a panel of amino acids, including lysine, as potential tumor surgical margin biomarkers for OSCC [5]. Notably, the hydroxylation of lysine is a key step for collagen cross-linking and deposition, which can lead to many collagens related diseases, such as fibrosis and cancer [6,7]. Collagen is a large family of extracellular matrix proteins and is the most abundant protein in vertebrates. Type I collagen, the predominant genetic type in the collagen family, is the major fibrillar component in most connective tissues [8]. Procollagen-lysine, 2-oxoglutarate 5-dioxygenase 2 (PLOD2), also known as lysyl hydroxylase 2 (LH2), is a key enzyme that mediate the formation of stable collagen cross-links by catalyzing the hydroxylation of lysine [9,10]. Accumulating evidence indicates that PLOD2 is significantly overexpressed and related to poor prognosis of patients in several types of solid tumors, such as hepatocellular carcinoma [11], pancreatic cancer [12], sarcoma [13], and head and neck squamous cell carcinoma (HNSCC) [14]. Nevertheless, little is known about the specific spatial distribution of PLOD2 in different cell types in tumor microenvironment (TME). Recently, although some articles also report that there are increased expressions of PLOD2 in OSCC cell lines [15,16]. Saito et al. demonstrated an aberrant collagen-modifying enzymes including PLOD2 in OSCC, but with a small sample size of 15 [17]. Thus, a more extensive analysis in OSCC patients is needed to validate its spatial expression profiles and clinical significance.

Therefore, for the first time, we detected the spatial expression patterns of PLOD2 in tumor cells (TCs), fibroblast-like cells (FLCs) and tumor-infiltrating lymphocytes (TILs) in a cohort of 100 OSCC patients. Moreover, we evaluated the correlation between PLOD2 and patients’ clinicopathological features and analyzed its diagnostic and prognostic value. Lastly, gene set enrichment analysis (GSEA) was performed to understand the HALLMARK pathway difference between high and low PLOD2 expression groups.

## 2. Materials and Methods

### 2.1. Patients and Samples

All methods used for this study were approved by the Ethics Committee of Nanjing Stomatology Hospital. The study was conducted in accordance with the Declaration of Helsinki. A total of 128 OSCC patients were recruited in the study. None of the patients received preoperative chemotherapy, radiotherapy, or other cancer-related treatments. Patients with history of systemic illness or missing survival data were excluded. These patients with primary tumor were diagnosed by hematoxylin and eosin staining by two experienced pathologists. Eight cases were recruited for gas chromatography–mass spectrometry (GC–MS)-based untargeted metabolic analysis, 20 cases were used for ultra–high performance liquid chromatography-tandem mass spectrometer (UHPLC-MS/MS) targeted metabolomics for quantitative analysis. And among these 28 patients which 18 cases were used for western blot analysis, respectively. These 28 cases (including tumor tissues and matched nontumor tissues from the surgical margin) were collected within 30 min after surgical resection, frozen in liquid nitrogen and stored at −80 °C before use. The other 100 cases were used for immunohistochemistry (IHC) study. These 100 patients included 21 cases of gingival cancer, 23 cases of buccal cancer, 10 cases of palate cancer and 46 cases of tongue cancer. They were followed up for 3–68 months, and the median was 38 months. The inclusion and exclusion criteria of patients were the same as those of our previous studies [18]. Written informed consents were obtained from all the patients. All these primary OSCC patients were diagnosed by two experienced pathologists using hematoxylin and eosin staining.

### 2.2. GC–MS-Based Untargeted Analysis

In order to explore reliable biomarkers of OSCC, we used GC–MS-based untargeted analysis to find the difference of potential metabolite between tumor and matched adjacent normal tissues of 8 OSCC patients. Materials and reagents for GC–MS analysis were prepared according to previous method [5]. GC–MS analysis was performed by Trace 1310 Gas Chromatograph equipped with an AS 1310 auto sampler, which connected a TSQ 8000 triple quadrupole mass spectrometer (Thermo Scientific, Waltham, MA, USA).

### 2.3. UHPLC-MS/MS Targeted Quantitative Analysis

To verify the previous results, we increased the sample size and measured the amount of 9 most frequently elevated amino acids in 20 OSCC patients using UHPLC-MS/MS-based targeted quantitative analysis. An Agilent 1290 Infinity II series UHPLC System (Agilent Technologies, Santa Clara, CA, 95051, United States), equipped with a Water ACQUITY UPLC BEH Amide column (100 × 2.1 mm, 1.7 μm) carried UHPLC separation. 1% formic acid in water construct mobile phase A, and 1% formic acid in acetonitrile construct mobile phase B. The temperature of the column was set to 35 °C. The temperature of auto-sample temperature was set to 4 °C and the injection volume was set to 1μL. We applied Agilent 6460 triple quadrupole mass spectrometer (Agilent Technologies, Santa Clara, CA, 95051, USA) equipped with an AJS electrospray ionization (AJS-ESI) interface for metabolite assay development. Through the standard of each metabolite, the optimal MRM parameters of the target metabolites are obtained. MRM data acquisition and processing were performed by Agilent Mass Hunter Work Station Software (B.08.00, Agilent Technologies, Santa Clara, CA, 95051, USA).

### 2.4. Public Databases Analysis

Since PLOD2 is a key enzyme that catalyzes the hydroxylation of lysine, we wanted to tell whether the elevated lysine in OSCC was associated with the abnormal expression of PLOD2. Thus, we firstly analyzed the differential mRNA expression level of PLOD2 between tumor and paired normal tissues in HNSCC using different public databases. Data were extracted from Oncomine database (http://www.oncomine.org/, accessed on 21 August 21) to study the expression of PLOD2 mRNA level in HNSCC. We also used the Gene_DE module of TIMER2 database (http://timer.comp-genomics.org/, accessed on 21 August 21) to evaluate the differential expression between tumor and normal tissues for PLOD2 across all The Cancer Genome Atlas (TCGA) tumors.

### 2.5. Western Blot Assay

We next detected the protein levels of PLOD2 in 18 paired samples collected from primary OSCC patients in our hospital. Twenty milligrams of OSCC patients’ tissues (tumor or paired normal tissues) and 200 ul Radio-Immunoprecipitation Assay (RIPA) lysis buffer with a mixture of protease and phosphatase inhibitors were placed in a tissue grinder for protein extraction. Equal amounts of proteins were separated through sodium dodecyl sulfate polyacrylamide gel electrophoresis (SDS-PAGE) and blocked in 3% Bovine Serum Albumin (BSA). After incubation with anti-PLOD2 primary antibody (1:1000 dilution; 21214-1-AP, Proteintech) at 4 °C overnight and Horseradish Peroxidase (HRP) conjugated secondary antibody for 1 h, protein bands were detected by the protein imaging system (Tanon5200). Immunoblots for the presented data were analyzed from at least 3 experimental repeats.

### 2.6. IHC Analysis

Given that till now the spatial distribution of PLOD2 in OSCC tissues is not known, we further performed IHC analysis to investigate the expression profile of PLOD2 on different cell types, including TCs, FLCs and TILs in 100 OSCC patients. IHC was performed on 2 μm formalin-fixed paraffin-embedded sections using anti-PLOD2 (21214-1-AP; Proteintech). Slides were deparaffinized with xylene and rehydrated in an ethanol series. Antigen retrieval was performed with TE buffer (pH 9.0) in a pressure cooker. Then, endogenous peroxidase activity was blocked with a 3% hydrogen peroxide solution. After washing in phosphate-buffered saline (PBS; pH 7.4) for three times, slides were incubated with primary antibody against PLOD2 (1:200 dilution; 21214-1-AP, Proteintech) at 4 °C overnight. Polink-2 plus HRP Detection Kit was used as the secondary antibody at 37 °C for 40 min. Finally, slides were developed in diaminobenzidine (DAB). The primary antibody was replaced by rabbit IgG as negative control.

### 2.7. Quantification of IHC

The expression patterns of PLOD2 in TCs, FLCs and TILs were visualized and scored by two independent pathologists who were blinded to the clinical data, and the mean value was adopted for further analysis. The scoring details of IHC was performed as previously described [19]. The percentage score of PLOD2 was defined as 0, <5%; 1.6–25%; 2, 26–50%; 3, 51–75%; and 4, >75% positive cells. Staining intensity score was defined as 0, negative staining; 1, weak staining; 2, moderate staining; and 3, strong staining. Final immunohistochemical scores were calculated by multiplying the percentage score combined with the intensity score. The expression level of PLOD2 was defined as “Low” when under the median value and as “High” when equal to or greater than the median value.

### 2.8. Gene Set Enrichment Analysis

To identify the potential signaling pathways involved in PLOD2 regulation in HNSCC, GSEA was performed using HNSCC data from TCGA database. GSEA software v4.1.0 provided by Broad Institute (https://www.gsea-msigdb.org/gsea/index.jsp/, accessed on 8 September 21) was utilized to perform GSEA using Head and Neck Squamous Cell Carcinoma (TCGA, Firehose Legacy) gene set download from cBioPortal (http://www.cbioportal.org/, accessed on 8 September 21). Gene sets with |NES| > 1, *p*-value < 0.05 and FDR < 0.25, after 1000 permutations, were taken into the count as significantly enriched gene sets.

### 2.9. Statistical Analysis

SPSS 22.0 software (SPSS Inc) and GraphPad Prism 8.0 were used for data analysis and figure process. The results of the experiments were presented as mean ± SEM. We evaluated the correlation between PLOD2 expression and clinicopathological characteristics of patients with OSCC through Pearson’s chi-square test. We further analyzed whether increased PLOD2 was related to postoperative recurrence as well as metastasis by the Mann-Whitney U test. To confirm the prognostic significance of PLOD2 for OSCC, we performed KM survival analysis to analyze the survival of recruited patients [20]. Survival analysis including overall survival (OS), disease-free survival (DFS), recurrence-free survival (RFS) and metastasis-free survival (MFS) were evaluated by Kaplan-Meier (KM) and log-rank test. Further univariate and multivariate analysis were performed by Cox proportional hazards regression model to identify the risk factors for OSCC with adjusted hazard ratio (HR) and 95% confidence interval (CI). All the analyses were two-sided test and considered statistically significant at a *p* < 0.05 level.

## 3. Results

### 3.1. Lysine was Significantly Elevated in OSCC Tissues

GC–MS-based untargeted analysis showed that 16 out of 244 amino acids were most significantly elevated in OSCC tumor tissues (fold change >1.5 and *p* < 0.05), including lysine (Figure 1a). To verify this result, we increased the sample size and measured the amount of 9 most frequently elevated amino acids in 20 OSCC patients using UHPLC-MS/MS-based targeted quantitative analysis. Receiver operating characteristics (ROC) curve analysis showed that lysine could most effectively distinguish tumor from normal tissues, with the area under the curve (AUC) 0.859 (*p* = 0.0035, Figure 1b,c). The concentrations of lysine in tumor and normal tissues were presented in Figure 1d.

### 3.2. PLOD2 mRNA and Protein Levels Were Upregulated in OSCC Patients

Data from Oncomine and TIMER2 both demonstrated that compared with normal tissues, PLOD2 mRNA level was significantly upregulated in tumor tissues in HNSCC patients (Figure 2a,b). Western blot analysis proved that PLOD2 protein level was indeed elevated in OSCC tumor tissues (*p* < 0.001, Figure 2c).

### 3.3. Spatial Distribution of PLOD2 and Its Correlation with Clinicopathological Characteristics of OSCC Patients

We found that PLOD2 was ubiquitously expressed in TCs and FLCs but absent in TILs (Figure 3a). Representative IHC images of low and high expression of PLOD2 are presented in Figure 3b. Pearson’s chi-squared test showed that expression of PLOD2 had no significant relationship with gender, age, smoking habits, or T stage. However, upregulated PLOD2 in FLCs (PLOD2^FLCs^) were positively correlated with worse tumor differentiation (*p* < 0.001), worst pattern of invasion (WPOI) (*p* < 0.001) and high risk of lymph node metastasis (LNM) (*p* < 0.001) (Table 1 and Figure 4a,c). Additionally, highly expressed PLOD2 in TCs (PLOD2^TCs^) were closely related to the late differentiation (*p* = 0.034) and deterioration of WPOI (*p* < 0.001) (Figure 4d,e).

### 3.4. Highly Expressed PLOD2 Was Associated with Higher Postoperative Metastasis Risk and Poor Survival Time

The Mann-Whitney U test results showed that the highly expressed PLOD2^FLCs^ and PLOD2^TCs^ had a higher risk of postoperative metastasis (*p* < 0.05, Figure 4f) but had no significant correlation with recurrence (*p* > 0.05, Figure 4g). Kaplan-Meier (KM) and log-rank test demonstrated that patients with higher PLOD2^FLCs^ and PLOD2^TCs^ had shorter OS, RFS, MFS and (DFS (*p* < 0.001, Figure 5a–h). Similarly, KM survival analysis using TCGA database also revealed that high expression of PLOD2 had poor OS (*p* = 0.0184) and DFS (*p* = 0.0192) in HNSCC patients (Figure 5i–j).

### 3.5. PLOD2^FLCs^ Was an Independent Prognostic Factor for OSCC

Univariate analyses shown in Table 2 and Table 3 revealed that gender, differentiation, WPOI, PLOD2^TCs^ and PLOD2^FLCs^ presented to be risk factors for OS, DFS, RFS and MFS in OSCC. However, multivariate analyses indicated that only PLOD2^FLCs^, but not PLOD2^TCs^, was an independent risk factor for OS (HR = 6.127, 95% CI = 2.046–18.347, *p* = 0.001 vs. 2.202, 0.975–4.974, 0.057), DFS (HR= 6.425, 95% CI = 2.068–19.267, *p* = 0.002 vs. 2.065, 0.954–4.769, 0.068), RFS (HR = 6.024, 95% CI = 1.975–18.373, *p* = 0.002 vs. 1.972, 0.847–4.593, 0.115), MFS (HR = 6.796, 95% CI = 2.244–20.582, *p* = 0.001 vs. 2.222, 0.990–4.986, 0.057) in OSCC (Table 2 and Table 3).

### 3.6. Signaling Pathways Involved in PLOD2 Regulation in HNSCC

As shown in Table 4, the top 20 most enriched signaling pathways or biological processes according to the NES score permutation have been previously characterized. As And the top 8 signaling pathways with most significantly enriched in hallmark, including epithelial–mesenchymal transition, mitotic spindle, G2M checkpoint, TGF-beta signaling, hypoxia, angiogenesis, protein secretion, and UV response were shown in Figure 6. This section may be divided by subheadings. It should provide a concise and precise description of the experimental results, their interpretation, as well as the experimental conclusions that can be drawn.

## 4. Discussion

PLOD2 is significantly overexpressed in several malignant tumors but no studies have been performed on what the cell types express this protein [13,21,22,23,24]. To our knowledge, our study is the first to assess the spatial expression pattern of PLOD2 in different major cell types in the TME of OSCCs. We demonstrated that PLOD2 was frequently expressed in TCs and FLCs but absent in TILs in OSCC. We also confirmed that upregulated PLOD2 in FLCs and TCs both had worse differentiation and WPOI, and correlated with increased risk of metastasis. In addition, patients with high PLOD2^FLCs^ were found to have greater LNM probability. PLOD2^FLCs^, rather than PLOD2^TCs^, was an independent risk factor for OS, DFS, RFS and MFS in OSCC (Table 2 and Table 3). This is different from that upregulation of the total PLOD2 is considered as an independent factor for poor prognosis in hepatocellular carcinoma [10]. Since proteins in different cells may play various roles, we believe it is crucial to elucidate the expression profile of PLOD2 in OSCC to benefit cell targeted research in the future.

It is well known that FLCs, the most abundant mesenchymal cells in TME, play a vital role in the formation of collagen fibers and associated with tumor metastasis [25,26]. FLCs can recombine collagen fibers, which hindered the migration of tumor cells through proteolysis and matrix remodeling and produced the trajectory of tumor cell [27]. According to the literatures, PLOD2 can be observed in cancer-associated fibroblasts in melanoma, lung adenocarcinoma, and liver cancer [28,29]. In addition, FLCs induce collagen cross-linking switch in tumor stroma through PLOD2, thus affecting the invasiveness of tumor cells [30]. In this study, we showed that PLOD2 was most widely expressed in FILs of OSCC patients, which indicates the importance of PLOD2FLCs in the progression and metastasis of OSCC.

Accumulating evidence has suggested that PLOD2 has the effect of promoting tumor progression in various solid tumors. For example, in esophageal squamous cell carcinoma, the high expression of PLOD2 was closely related to tumor size and intrahepatic metastasis [24]. In colorectal cancer, PLOD2 was closely related to tumor progression, grading and N stage [31]. In cervical cancer, PLOD2 influenced migration and invasion of cervical cancer cells [32]. Saito T et al. showed that PLOD2 levels in OSCCs were significantly correlated with the advanced cancer stages and presence of regional lymph node metastasis [17]. Similarly, our research showed that OSCC patients with higher PLOD2 in FLCs had worse tumor differentiation and WPOI and higher risk of lymph node metastasis compared with lower PLOD2^FLCs^. Upregulation of PLOD2 in TCs was significantly associated with the late differentiation and deterioration of WPOI, a histologic risk assessment score that represents invasive activity of tumors. And what’s more, highly expressed PLOD2 both in TCs and FLCs were associated with higher postoperative metastasis risk (Figure 4).

We also used bioinformatic analysis to confirm our results. Since OSCC is the most common type of epithelial malignant tumor in the head and neck, using the HNSCC cases in TCGA database for OSCC study is frequently applied by researchers [17,33]. In this study, by using the HNSCC cases in TCGA database, we found significant increases of PLOD2 mRNA levels in tumor tissues compared with normal tissues (Figure 2), and highly expressed PLOD2 was associated with poor OS and DFS (Figure 5i,j). Our IHC analyses are consistent with these results that poor OS, RFS, MFS and DFS are associated with the increased expression of PLOD2 in OSCC (Figure 5a–h). In other solid tumors, such as hepatocellular carcinoma [34] and breast cancer [35], high expression of PLOD2 is also reported to be significantly related to the decrease of DFS.

Interestingly, we found that EMT, TGF-β, and hypoxia gene sets were the most significantly enriched genes in higher PLOD2 expression groups in HNSCC cases (Figure 6). Evidence has emerged that PLOD2 is involved in the regulation of extra matrix collagen and tumor metastasis through EMT, TGF-β, and hypoxia signals. For instance, in HIF-1α-deficient human sarcoma, the expression of ectopic PLOD2 restored the potential of migration and metastasis, while inhibiting the activity of PLOD2 resulted in decreased tumor metastasis [13]. In endometrial carcinoma cells, PLOD2 modulated cell migration, invasion, and EMT via PI3K/Akt signaling under hypoxic conditions [36]. These indicate that EMT, TGF-β and hypoxia signals may contribute to poor clinical outcomes such as metastasis with PLOD2 overexpression in OSCC patients.

Taken together, we identified and validated PLOD2 as an adverse prognostic biomarker for OSCC. Thus, PLOD2 in FLCs and TCs may be a promising therapeutic target for primary OSCC. Altogether, the study mainly explored the prognostic value of PLOD2 in OSCC. However, the molecular function and regulation pathway of PLOD2 in tumorigenesis of OSCC remained unexplored.

## Figures and Tables

**Figure 1 biomolecules-11-01842-f001:**
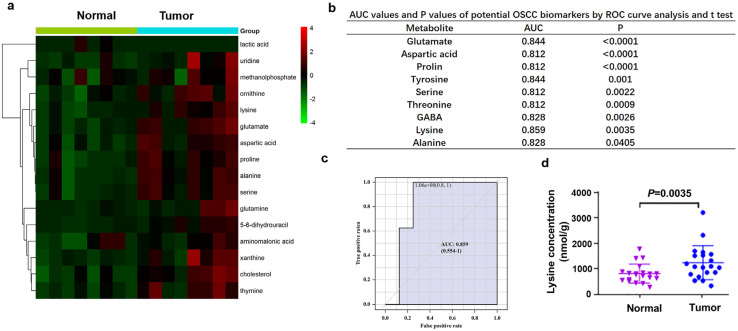
Differential lysine levels in tumor and adjacent normal tissues of oral squamous cell carcinoma (OSCC). (**a**) Heat map of differential amino acids in tumor and paired normal tissues of 8 OSCC patients. (**b**) Nine amino acids significantly up–regulated in OSCC tumor tissues compared with matched normal tissues using UHPLC–MS/MS–based quantitative analysis (*n* = 20). (**c**) ROC curve and fold change of lysine between tumor and normal group (AUC = 0.859, *p* = 0.0035). (**d**) The level of lysine between tumor and normal tissues of OSCC patients (*n* = 20).

**Figure 2 biomolecules-11-01842-f002:**
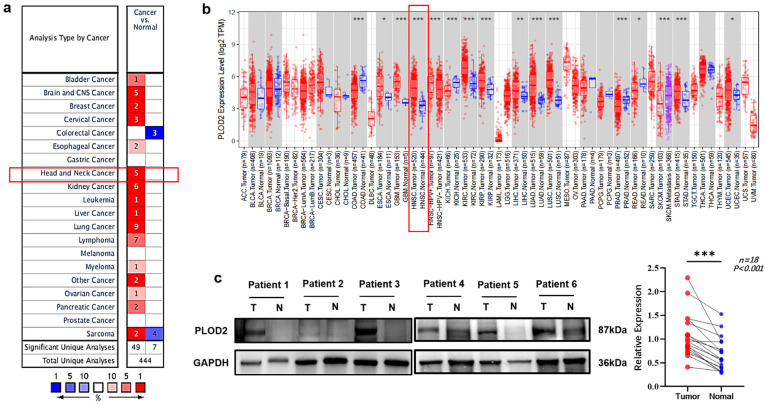
The expression level of PLOD2 in tumor and normal tissues of HNSCC and OSCC. (**a**) PLOD2 expression in HNSCC using Oncomine database, cell color is determined by the best gene rank percentile for the analyses within the cell. (**b**) PLOD2 expression levels in all tumors and adjacent normal tissues across TIMER2 online database. (**c**) The difference in PLOD2 protein expression levels between OSCC tumor tissues and normal controls was investigated by western blot. * *p* < 0.05, ** *p* < 0.01, *** *p* < 0.001 compared to normal tissues.

**Figure 3 biomolecules-11-01842-f003:**
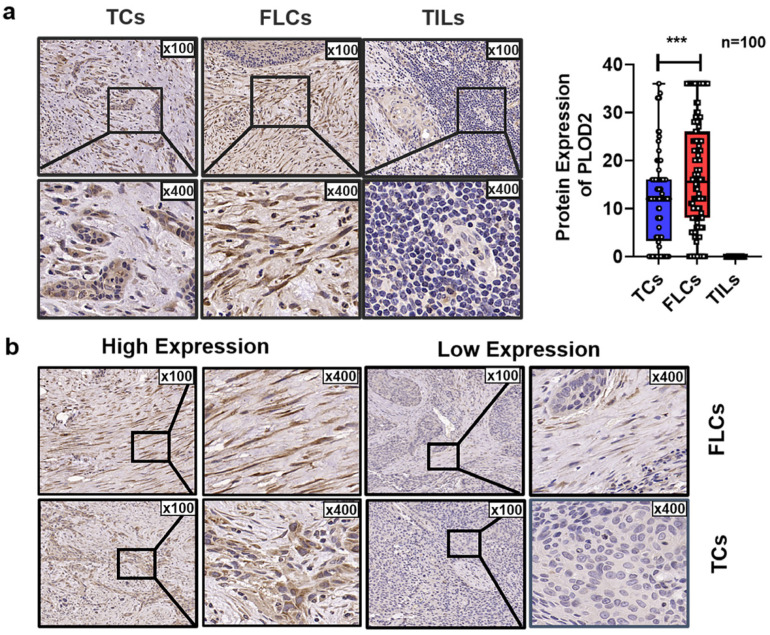
PLOD2 spatial expression in different cell types of OSCC patients. (**a**) Representative immunohistochemistry (IHC) staining of PLOD2 in tumor cells (TCs), fibroblast-like cells (FLCs), and tumor-infiltrating lymphocytes (TILs). (**b**) Representative high and low expression of PLOD2 in FLCs and TCs. *** *p* < 0.001.

**Figure 4 biomolecules-11-01842-f004:**
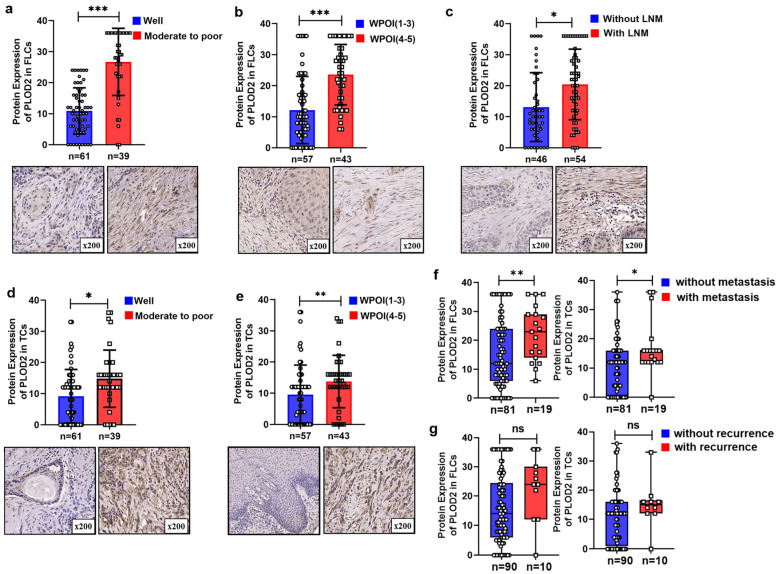
PLOD2 expression with different clinical characteristics. (**a**–**c**) PLOD2 with different differentiations, WPOI and lymph node metastasis in FLCs. (**d**,**e**) PLOD2 with different differentiations and WPOI in TCs. Correlation between PLOD2 expression and metastasis (**f**) or recurrence (**g**) status in TCs and TILs. * *p* < 0.05, ** *p* < 0.01, *** *p* < 0.001.

**Figure 5 biomolecules-11-01842-f005:**
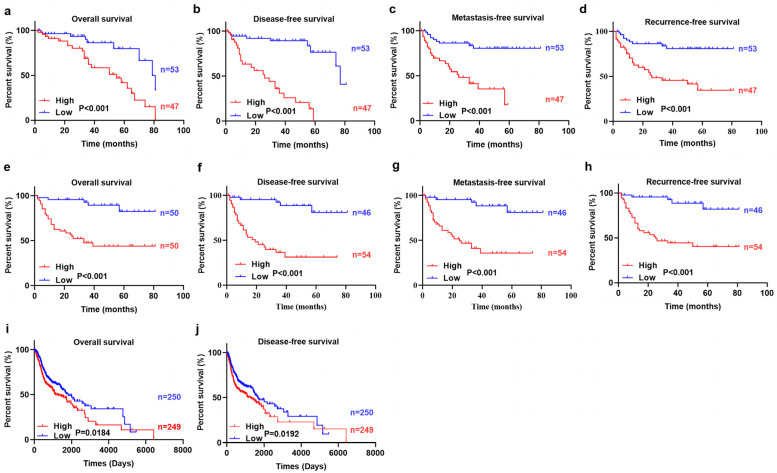
Survival analysis results. KM survival curves for OS, DFS, MFS, and RFS of OSCC patients according to the expression of PLOD2 in TCs (**a**–**d**) and FLCs (**e**–**h**), respectively. TCGA database was used to evaluate the relationships between PLOD2 expression and patient survival. High expression of PLOD2 has poor OS (**i**) and DFS (**j**) in HNSCC patients.

**Figure 6 biomolecules-11-01842-f006:**
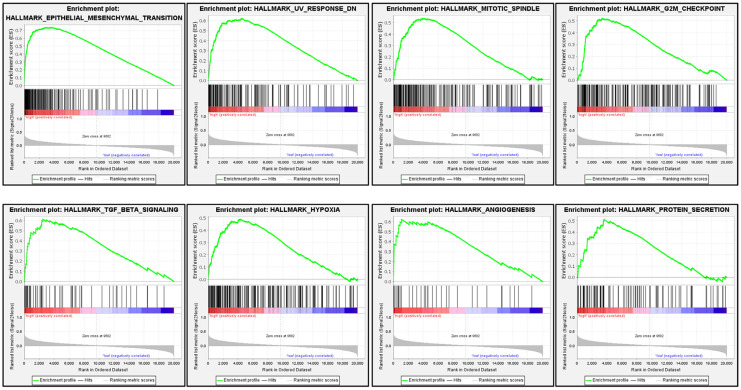
GSEA plots of PLOD2 ranked in the top 8 associated with signaling pathways in hallmark datasets. The red color indicates a strong correlation, and the blue color indicates a negative correlation.

**Table 1 biomolecules-11-01842-t001:** Association between PLOD2 expression and clinicopathological characteristics in OSCC patients.

		FILs					TCs			
Characteristics	N	Low (%)	High (%)	χ^2^	*p* Value	N	Low (%)	High (%)	χ^2^	*p* Value
**Gender**										
Male	58	31 (53.4%)	27 (46.6%)	0.369	0.543	58	32 (51.7%)	26 (48.3%)	0.095	0.758
Female	42	19 (45.2%)	23 (54.8%)			42	21 (65.9%)	21 (34.1%)		
**Age (year)**										
<60	43	22 (51.2%)	21 (48.8%)	0	1	43	26 (58.1%)	17 (41.9%)	1.203	0.273
≥60	57	28 (49.1%	29 (50.9%)			57	27 (57.1%)	30 (42.9%)		
**Smoking**										
NO	60	26 (43.3%)	34 (56.7%)	2.042	0.153	60	27 (50.8%)	33 (49.1%)	3.093	0.079
YES	40	24 (60%)	16 (40%)			40	19 (57.5%)	22 (42.5%)		
**T stage**										
Ⅰ-Ⅱ	59	31 (52.5%)	28 (47.5%)	0.165	0.684	59	34 (57.6%)	25 (42.4%)	0.825	0.364
Ⅲ-Ⅳ	41	19 (46.3%)	22 (53.7%)			41	19 (57.5%)	22 (42.5%)		
**LNM**										
NO	46	34 (73.9%)	12 (26.1%)	17.754	**<0.001 ***	46	28 (66.7%)	18 (33.3%)	1.573	0.21
YES	54	16 (29.6%)	38 (70.4%)			54	25 (50%)	29 (50%)		
**Differentiation**										
Well	61	41 (67.2%)	20 (32.8%)	16.814	**<0.001 ***	61	38 (67.8%)	23 (32.2%)	4.51	**0.034 ***
Moderate to poor	39	9 (23.1%)	30 (76.9%)			39	15 (42.5%)	24 (57.5%)		
**WPOI**										
1~3	57	40 (70.2%)	17 (29.8%)	19.747	**<0.001 ***	57	43 (75%)	14 (25%)	24.739	**<0.001 ***
4~5	43	10 (23.3%)	33 (76.7%)			43	10 (34.9%	33 (65.1%)		

TCs, tumor cells; FLCs, fibroblast-like cells; LNM, lymph node metastasis; WPOI, worst pattern of invasion; χ^2^, Pearson’s chi-squared test. * Represented those differences were considered statistically significant with *p* < 0.05.

**Table 2 biomolecules-11-01842-t002:** Cox-regression analysis of OS and DFS in OSCC patients.

Variables	Univariate Analysis		*p*	Multivariate Analysis		*p*
	HR	95%CI		HR	95%CI	
**OS**						
**Gender**: Male vs. Female	1.647	0.779–3.482	0.192			
**Age**: ≥60 vs. <60	2.194	1.014–4.745	**0.046 ***	2.219	1.005–4.903	**0.049 ***
**Smoking**: Yes vs. No	0.751	0.362–1.560	0.443			
**T Stage**: III-IV vs. I-II	0.881	0.430–1.802	0.728			
**LNM**: Yes vs. No	1.119	0.556–2.252	0.752			
**Differentiation**: Moderate to poor vs. Well	2.214	1.099–4.460	**0.026 ***	1.528	0.736–3.169	0.255
**WPOI**: 4–5 vs. 1–3	2.257	1.110–4.588	**0.024 ***	0.833	0.370–1.876	0.659
**Metastasis**: Yes vs. No	0.951	0.391–2.314	0.913			
**Recurrence**: Yes vs. No	0.862	0.262–2.839	0.807			
**PLOD2 in FLCs**: High vs. Low	8.131	2.841–23.273	**0.001 ***	6.127	2.046–18.347	**0.001 ***
**PLOD2 in TCs**: High vs. Low	3.141	1.514–6.520	**0.002 ***	2.202	0.975–4.974	0.057

**DFS**						
**Gender**: Male vs. Female	1.336	0.648–2.756	0.433			
**Age**: ≥60 vs. <60	2.121	1.006–4.715	**0.0048 ***	2.167	0.988–4.744	0.056
**Smoking**: Yes vs. No	0.604	0.292–1.250	0.174			
**T Stage**: III-IV vs. I-II	0.813	0.399–1.657	0.569			
**LNM**: Yes vs. No	1.158	0.582–2.301	0.676			
**Differentiation**: Moderate to poor vs. Well	2.289	1.136–4.554	**0.021 ***	1.367	0.685–2.868	0.45
**WPOI**: 4–5 vs. 1–3	2.912	1.435–6.134	**0.01 ***	1.168	0.548–2.865	0.654
**Metastasis**: Yes vs. No	1.428	0.712–3.865	0.189			
**Recurrence**: Yes vs. No	1.452	0.654–4.658	0.356			
**PLOD2 in FLCs**: High vs. Low	8.639	3.125–26.354	**0.001 ***	6.425	2.068–19.267	**0.002 ***
**PLOD2 in TCs**: High vs. Low	3.462	1.625–7.159	**0.002 ***	2.065	0.954–4.769	0.068

TCs, tumor cells; FLCs, fibroblast-like cells; OS, overall survival time; DFS, disease-free survival time; LNM, lymph node metastasis; WPOI, worst pattern of invasion; CI, confidence interval. * Represented those differences were considered statistically significant with *p* < 0.05.

**Table 3 biomolecules-11-01842-t003:** Cox-regression analysis of RS and MFS in OSCC patients.

Variables	Univariate Analysis		*p*	Multivariate Analysis		*p*
	HR	95%CI		HR	95%CI	
**RFS**						
**Gender**: Male vs. Female	1.703	0.805–3.604	0.164			
**Age**: ≥60 vs. <60	2.227	1.029–4.819	**0.042 ***	2.149	0.973–4.475	0.058
**Smoking**: Yes vs. No	0.684	0.329–1.420	0.308			
**T Stage**: III-IV vs. I-II	0.9	0.439–1.842	0.773			
**LNM**: Yes vs. No	1.183	0.588–2.380	0.638			
**Differentiation**: Moderate to poor vs. Well	2.269	1.126–4.574	**0.022 ***	1.438	0.693–2.987	0.33
**WPOI**: 4–5 vs. 1–3	2.547	1.253–5.180	**0.01 ***	0.943	0.409–2.173	0.891
**Metastasis**: Yes vs. No	0.881	0.362–2.141	0.779			
**Recurrence**: Yes vs. No	1.97	0.589–6.584	0.271			
**PLOD2 in FLCs**: High vs. Low	8.559	2.990–24.501	**0.001 ***	6.024	1.975–18.373	**0.002 ***
**PLOD2 in TCs**: High vs. Low	3.248	1.564–6.745	**0.002 ***	1.972	0.847–4.593	0.115

**MFS**						
**Gender**: Male vs. Female	1.448	0.686–3.059	0.332			
**Age**: ≥60 vs. <60	2.207	1.021–4.773	**0.044 ***	2.176	0.994–4.764	0.052
**Smoking**: Yes vs. No	0.721	0.347–1.500	0.381			
**T Stage**: III-IV vs. I-II	0.839	0.410–1.720	0.632			
**LNM**: Yes vs. No	1.175	0.583–2.366	0.652			
**Differentiation**: Moderate to poor vs. Well	2.298	1.141–4.626	**0.02 ***	1.287	0.609–2.720	0.5
**WPOI**: 4–5 vs. 1–3	3.41	1.622–7.168	**0.001 ***	1.365	0.593–3.143	0.465
**Metastasis**: Yes vs. No	2.326	0.910–5.944	0.078			
**Recurrence**: Yes vs. No	0.762	0.231–2.512	0.655			
**PLOD2 in FLCs**: High vs. Low	9.933	3.340–28.768	**0.001 ***	6.796	2.244–20.582	**0.001 ***
**PLOD2 in TCs**: High vs. Low	3.587	1.722–7.472	**0.001 ***	2.222	0.990–4.986	0.053

TCs, tumor cells; FLCs, fibroblast-like cells; RFS, recurrence-free survival time; MFS, metastasis-free survival; LNM, lymph node metastasis; WPOI, worst pattern of invasion; CI, confidence interval. * Represented those differences were considered statistically significant with *p* < 0.05.

**Table 4 biomolecules-11-01842-t004:** Gene sets enriched in the high PLOD2 expression phenotype.

TERM	ES	NES	NOM p-val	FDR q-val	FWER p-val
HALLMARK_EPITHELIAL_MESENCHYMAL_TRANSITION	0.74	3.51	<0.001	<0.001	<0.001
HALLMARK_UV_RESPONSE_DN	0.62	2.83	<0.001	<0.001	<0.001
HALLMARK_MITOTIC_SPINDLE	0.54	2.55	<0.001	<0.001	<0.001
HALLMARK_G2M_CHECKPOINT	0.52	2.47	<0.001	<0.001	<0.001
HALLMARK_TGF_BETA_SIGNALING	0.61	2.36	<0.001	<0.001	<0.001
HALLMARK_HYPOXIA	0.49	2.31	<0.001	<0.001	<0.001
HALLMARK_ANGIOGENESIS	0.62	2.21	<0.001	<0.001	<0.001
HALLMARK_PROTEIN_SECRETION	0.51	2.19	<0.001	<0.001	<0.001
HALLMARK_HEDGEHOG_SIGNALING	0.6	2.16	<0.001	<0.001	<0.001
HALLMARK_INFLAMMATORY_RESPONSE	0.45	2.13	<0.001	<0.001	<0.001
HALLMARK_E2F_TARGETS	0.44	2.11	<0.001	<0.001	<0.001
HALLMARK_KRAS_SIGNALING_UP	0.44	2.08	<0.001	<0.001	<0.001
HALLMARK_TNFA_SIGNALING_VIA_NFKB	0.42	2.01	<0.001	<0.001	<0.001
HALLMARK_ANDROGEN_RESPONSE	0.46	1.99	<0.001	<0.001	0.001
HALLMARK_IL6_JAK_STAT3_SIGNALING	0.45	1.91	<0.001	<0.001	0.003
HALLMARK_APICAL_JUNCTION	0.4	1.91	<0.001	<0.001	0.003
HALLMARK_GLYCOLYSIS	0.4	1.9	<0.001	<0.001	0.003
HALLMARK_COAGULATION	0.41	1.87	<0.001	<0.001	0.004
HALLMARK_IL2_STAT5_SIGNALING	0.36	1.7	<0.001	0.002	0.035
HALLMARK_COMPLEMENT	0.35	1.66	<0.001	0.003	0.054

ES, Enrichment score; NES, Normalized Enrichment Score; NOM p-val, Nominal *p* Value; FDR q-val, False Discovery Rate q Value; FWER p-val, Family-wise Error Rate *p* Value.

## Data Availability

Publicly available datasets were analyzed in this study. This data can be found here: https://gdc-hub.s3.us-east-1.amazonaws.com/download/TCGA-HNSC.htseq_counts.tsv.gz, accessed on 10 November 2021.

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
