# Peer review of "Identification and Validation of PLOD2 as an Adverse Prognostic Biomarker for Oral Squamous Cell Carcinoma"

_biomolecules, 2021, doi:10.3390/biom11121842_

Round 1

Reviewer 1 Report

Dear Editor,

thank you for your request to review the manuscript “Identification and validation of PLOD2 as an adverse prognostic biomarker for oral squamous cell carcinoma”.

The topic of the manuscript is interesting and pioneering. In my opinion, the paper should be accepted after making a revision.

I have  some comments:

- Firstly the material and methods part should be better organized while in statistical analysis and results many sentences should be move in a separate section. (I will better specify in my further comments)

- the Authors should improve the introduction by adding some information about fibrillar type1 collagen with a focus on cross linking.

- the Authors should improve Material and Methods and better specify the choice of the number of recruited cases  for GC-MS and UHPLC-MS/MS ( page 2 line 77-81)

- in the subsection 2.4 The Authors should move the sentences about statistical analysis (page 3 line 108-111)

-The Authors should better specify if they have performed the power-calculation of the study (section 2.9)

-the sections 3.1 ( line 161-163) 3.2 (line 181-184)  3.3 (201-203) and so on should be moved in Material and Methods in which The Authors should better explained

-In the  section  3 the Authors should be careful to write only information on the results

-  In the section clinical data and histopathological analysis the Authors should improve inclusion and exclusion criteria of  participants.

- in each table the Authors should specify the performed test and p-value.

- in the discussion the Authors should improve the first sentences 288-290 ( delete various articles … and begin with “PLOD2 is significantly overexpressed in several malignant tumors but no studies have been performed on what the cell types express…for example)

 -in the discussion the Authors should move the sentence the line 294-296 at the end of the introduction.

-The Authors should add the limits of the study after discussion section.

Author Response

Thank you very much for your comments to our manuscript entitled “Identification and validation of PLOD2 as an adverse prognostic biomarker for oral squamous cell carcinoma” (manuscript ID: biomolecules-1492056). We appreciate your valuable comments, which help us improve our manuscript quality.

Based on the comments we received, careful modifications have been made. All your comments, which were responded point by point, were listed at the uploaded Word file. Please see the attachment.

Reviewer 2 Report

It is a well-written article that focuses on the issue of PLOD2 as adverse  prognostic biomarker for OSCC patients. This is an important issue as several articles described the role of PLOD2 as biomarker for solid tumor in other anatomic sites. The methodology and statistical analysis is absolutely adequate.
In consideration of the potential clinical implications that could be realized by offering new strategies to the target therapy, I recommend the publication.

Author Response

Thank you for your commendation and this valuable opportunity of revision.

Reviewer 3 Report

Dear Authors

thank you for your paper. I've found it very well organized and written. Results are well reported in the text. The quality of the research is high. Images good. I want to suggest only to introduce in the text the number of patient/cases you studied, with the aim to better underline what rightly you stated in the text  "the PLOD2 may be a promising therapeutic target for OSCC", considering that a total of 128 OSCCs is a relevant number but further validation are obviously needed.

e.g. Identification and validation of PLOD2 as an adverse prognos-
tic biomarker for oral squamous cell carcinoma in a study group of 128 cases 

Thank you again for your paper.  

Author Response

Thank you very much for your comments to our manuscript entitled “Identification and validation of PLOD2 as an adverse prognostic biomarker for oral squamous cell carcinoma” (manuscript ID: biomolecules-1492056). We appreciate your valuable comments, which help us improve our manuscript quality.

Based on the comments we received, careful modifications have been made. All your comments, which were responded point by point, were listed at the uploaded Word file. Please see the attachment.

This manuscript is a resubmission of an earlier submission. The following is a list of the peer review reports and author responses from that submission.